# Characterizing primary transcriptional responses to short term heat shock in Down syndrome

Joseph F. Cardiello[1,5], Jessica Westfall[2], Robin Dowell[1,2,3,4], Mary Ann Allen●[1,3,4]*

1 BioFrontiers Institute, University of Colorado, Boulder, CO, United States of America, 2 Molecular, Cellular and Developmental Biology, University of Colorado, Boulder, CO, United States of America, 3 Linda Crnic Institute, University of Colorado, Denver, CO, United States of America, 4 BioFrontiers-Crnic Boulder Branch, University of Colorado, Boulder, CO, United States of America, 5 Department of Chemistry & Biochemistry, Colorado College, Colorado Springs, CO, United States of America

* mary.a.allen@colorado.edu

**Data Availability Statement:** Datasets generated during this study are deposited at the Gene Expression Omnibus (GEO): GSE173536, GSE173537.

## Abstract

Heat shock stress induces genome-wide changes in transcription regulation, activating a coordinated cellular response to enable survival. We noticed many heat shock genes are up-regulated in blood samples from individuals with trisomy 21. We characterized the immediate transcriptional response to heat shock of two lymphoblastoid cell lines derived from brothers with and without trisomy 21. The trisomy 21 cells displayed a more robust heat shock response after just one hour at 42˚C than the matched disomic cells.

## Introduction

An extra copy of chromosome 21 causes trisomy 21. Mostly, it is unclear how this extra copy of 1% of the genome leads to the phenotypes associated with trisomy 21. Trisomy 21 cells have an altered, often increased, response to key cellular perturbations. For instance, trisomy 21 cells show an increased interferon response relative to typical cells, likely driven by overexpression of four interferon receptor genes encoded on chromosome 21 [1–4]. Similarly, trisomic cells show signs of an elevated oxidative stress response, which may tie into the chromosome 21 encoded, oxidative stress-responsive NRF2 or SOD1 genes [5–7]. Yet altered cellular responses are not solely the result of the genes encoded on chromosome 21. Human aneuploids, regardless of the identity of the aneuploid chromosome, trigger a general stress response [8–12]. Therefore, we wished to understand how trisomy 21 cells respond to perturbations that do not have primary regulators on chromosome 21.

Heat shock is a potentially lethal stress that activates various cellular response processes, including the unfolded protein response. While three genes encoded on chromosome 21 are heat shock activated (HSF2BP, DNAJC28, HSPA13), none are known to be upstream regulators of the heat shock response. The major regulator of heat shock, HSF1, is a transcription factor located on chromosome 8. HSF1 is ubiquitously expressed and highly regulated through post-translational modifications, nuclear import, and protein interactions [13–16]. Following heat shock there is an increase in HSF1 DNA binding and HSF1 activity (reviewed in [13, 17]).

**Funding:** We would like to acknowledge funding from the Sie Foundation for funding JFC and MAA, and for funding from the R01HL156475 MAA and JW, and R01GM125871 for funding MAA and RDD. The funders had no role in study design, data collection and analysis, 480 decision to publish, or preparation of the manuscript.

**Competing interests:** RDD and MAA have a patent for "Methods for predicting transcription factor activity" that is not directly related to the work contained in this data note.

Activation of HSF1 results in genome wide transcription changes including activation of the production of heat shock proteins, and repression of thousands of genes [18, 19].

In the case of trisomy 21, there are mixed reports concerning the impact of heat shock. Aneuploidy in yeast cells leads to increased cell to cell variation in response to heat shock [20]. In human fibroblasts, trisomy 21 cells failed to properly activate a couple of key heat shock proteins after heat shock [21]. In untreated clinical blood samples of a cohort of individuals with trisomy 21 (Fig 2), we found an irregular elevation of heat shock target genes in individuals with trisomy 21.

This lack of clarity led us to use heat shock to investigate how trisomy 21 cells mount a robust heat shock stress response (Fig 1). Therefore, we examined the primary effects of heat shock on lymphoblastoid cell lines from two brothers. We found, by multiple omics assays, that after a short, mild heat shock stress, the trisomy 21 lymphoblastoid cell line activates primary HSF1-regulated transcriptional responses more robustly than the diploid cells.

## Materials and methods

### Data from the Human Trisome project

Graphs (Fig 2) were created using the Human Trisome project interactive website data. We used the Somascan platform, the Kolmogorov-Smirnov Test, and the Benjamin-Hochberg adjustment method for the protein graphs. For the transcriptome data, we used the Linear model with the the Benjamin-Hochberg adjustment method.

### Cell lines

The lymphoblastoid cell lines of two individuals were obtained from the Translational Nexus Biobank (COMIRB 08–1276), University of Colorado School of Medicine, JFK Partners. The Translational Nexus Biobank acts as the honest broker for the samples used for this study and samples were provided to the study team in a deidentified manner. As such, this study is considered non-human subjects research, and additional IRB approval was not required. At the Biobank lymphoblastoid cell lines were derived from blood samples acquired from two brothers of similar age, one with and one without trisomy 21. These cell lines were produced by infecting blood samples from two brothers with Epstein Barr virus. The two sibling cell lines were immortalized with the same batch of EBV preparation. The authors had no access to information that could identify individual participants during or after data collection.

We chose to conduct this study in cell lines derived from brothers in order to reduce the effects of developmental, disease-related, and genetic differences on background differences in the cellular state. Most importantly, working with cell lines meant that we could challenge the cells with heat shock and assess the primary effects of this stress on gene expression in each cell line, enabling us to compare the state of chromatin accessibility, transcription, and transcript levels after heat shock back to the steady state levels in each respective cell line.

Lymphoblastoid cultures were grown in RPMI media supplemented with 20% FBS, 100 U/ml penicillin, 100 ug/ml streptomycin, and 2mM L-glutamine. Cell cultures were maintained in incubators with 5% $CO_2$ at 37˚. For the heat shock, flasks of cells were suspended in water baths at 42˚ for 1 hour. Control condition cell flasks were maintained in incubators prior to harvesting. Because control flasks were maintained in the incubator, there may also be a difference in $CO_2$ consumption in the 42˚ samples. Each of the replicates was grown on different days.

Drosophila S2 cells were used as spike-ins. They were grown in Schneider's Drosophila medium with 10%FBS at 25˚C. Drosophila S2 cell nuclei were prepared in one batch and spiked into each nuclei prep at approximately 1%.

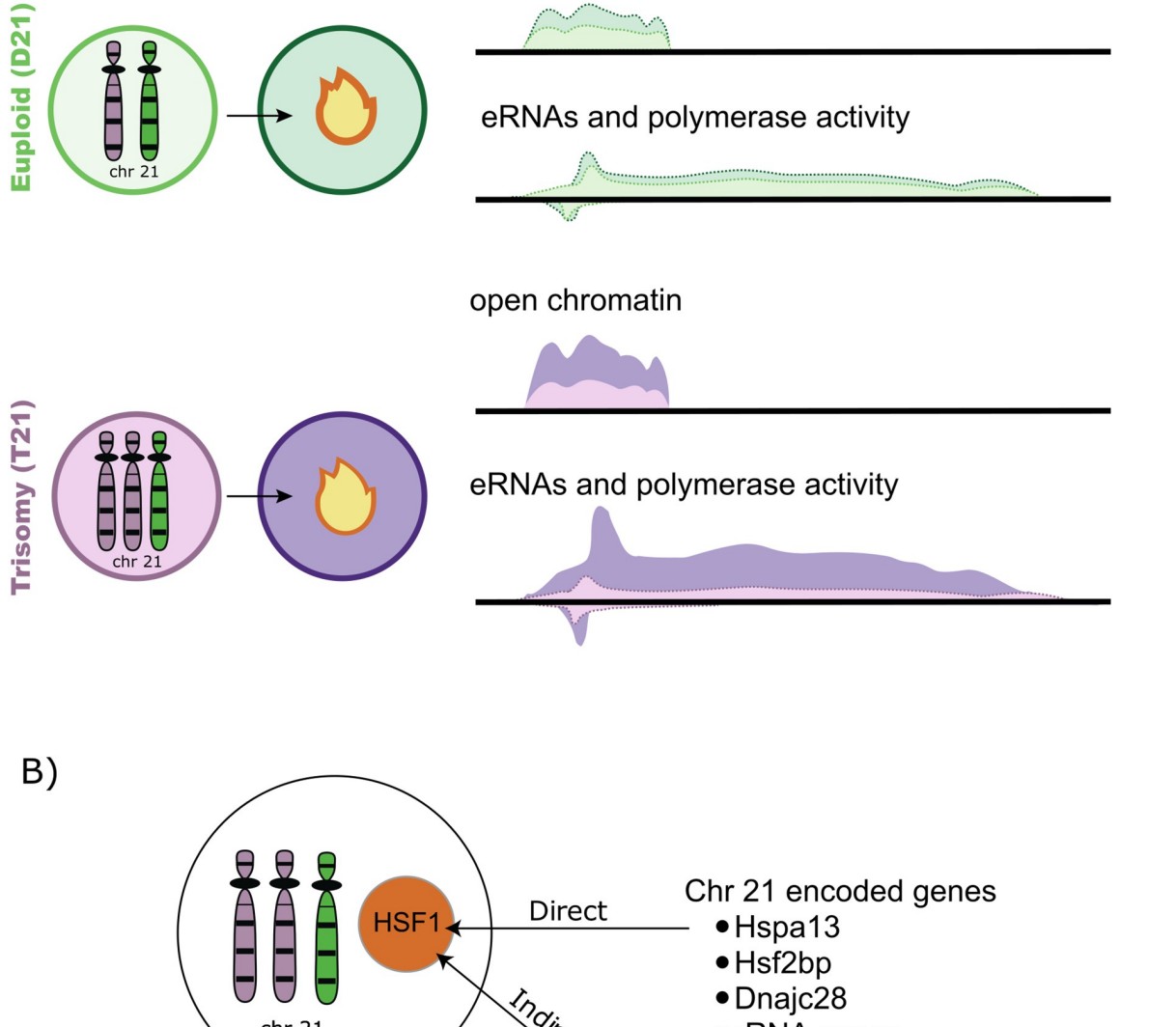

**Fig 1. Graphical abstract.** A) Cells with trisomy 21 have increased acute heat shock activated HSF1 transcription factor function as determined by both PRO-seq and ATAC-seq analysis. B) Several trisomy 21 cellular changes may contribute to an increased response to heat shock.

## PRO-seq experiments

**Nuclei isolations.** Cells were grown on in suspension in T75 flasks. Each of the two replicates was grown on different days. After cell treatments, aliquots of cells in media were washed 3X with cold PBS (phosphate buffer saline). Cells were next incubated on ice with

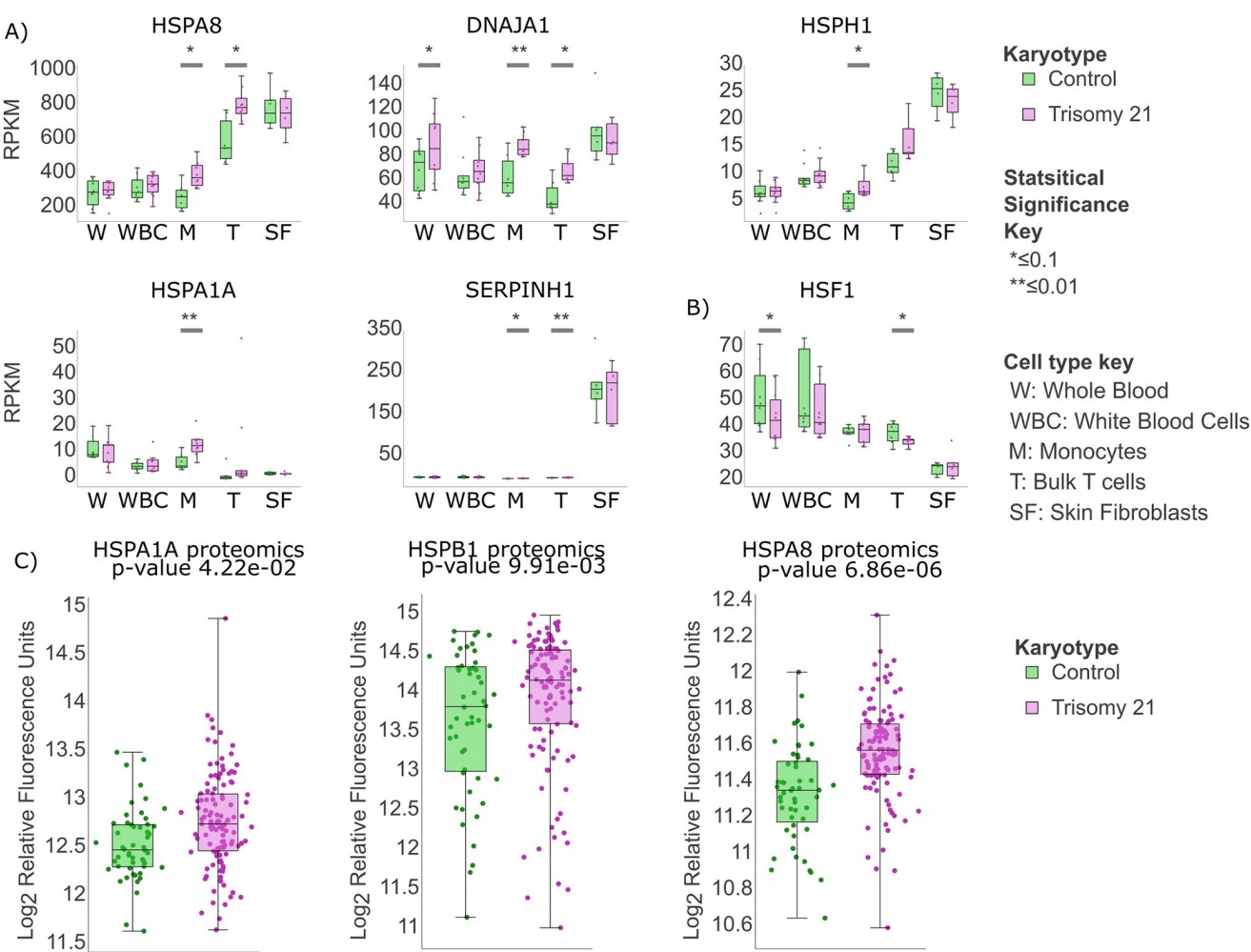

**Fig 2. Individuals with trisomy 21 have elevated levels of some heat shock-regulated genes under normal conditions.** All Data from the Human Trisome project [36]. A) Several heat shock genes (HSPA8, DNAJA1, HSPH1, HSPA1A, SERPINH1) are differentially expressed (RNA-seq) in some blood cell lineages in individuals with trisomy 21 (purple) compared to disomic controls (green). B) HSF1 is different, but often down, in trisomy 21. Multiple clinical samples shown: whole blood (W), white blood cells (WBC), monocytes (M), bulk T cells (T), and skin fibroblasts (SF). Significance key: * ≤ 0.1, ** ≤ 0.01. C) Clinical blood sample proteomics data [37] shows elevated levels for some heat shock induced genes (HSPA1A, HSPB1, and HSPA8) in people with trisomy 21.

buffer 1 containing 10 mM Tris pH 7.5, 2mM MgCl2, and 3 mM CaCl2 in DEPC (1% diethylpyrocarbonate treated) water, before centrifuging for 10 minutes at 1000 x G at 4˚C. Buffer was then aspirated and cells were resuspended in 1ml of buffer 2 containing 10 mM Tris pH 7.5, 2mM MgCl2, and 3 mM CaCl$_2$, 0.5% IGEPAL (octylphenoxy poly(ethyleneoxy) ethanol, branched, a nontoxic NP40 alternative), 10% glycerol, 2U/ml of SUPERase-IN, 1X protease inhibitor cocktail (Roche), and 1mM DTT (dithiothreitol) in DEPC treated water. After resuspension, 9ml more of buffer 2 was added, samples were mixed by inverting the tube, and samples were then centrifuged for 10 minutes at 1000xG at 4˚C. Solution was then aspirated, leaving behind cell nuclei. Nuclei were cleaned up by repeating the process of resuspending in 1ml of buffer 2, bringing solution up to 10ml total buffer 2, spinning samples at 1000 x G for 1 minutes, and aspirating media. White nuclei pellets were then re-

suspended in 1ml of buffer 3 containing 50 mM Tris pH 8.3, 5 mM $MgCl_2$, 40% glycerol, and 0.1 mM EDTA (ethylenediaminetetraacetic acid) pH 8.0 in DEPC treated water. Nuclei in buffer 3 were transferred to pre-lubricated 1.7ml tubes, centrifuged for 5 minutes at 600 x G at 4˚C, and aspirated. Nuclei pellets were then re-suspended in 500 ul of buffer 3, the centrifugation process was repeated, and final nuclei pellets were suspended in 110 ul buffer 3. A 1 ul aliquot of nuclei suspensions was diluted in PBS for nuclei quantification. Samples were diluted to make 100ul aliquots of 10 million nuclei each, and then flash frozen in liquid nitrogen and stored at -70˚C.

**Nuclei run-on and RNA preparation.** Two replicates of the 4 sample/condition PRO-seq experiments were generated on different batches of cells on different days. Precision run-on sequencing experiments (PRO-seq) were carried out according to the PRO-seq protocol [19]. 10 million nuclei were used per sample. Before the second replicate, before the initial cell pellet collection by centrifugation, frozen drosophila S2 cells were spiked into each sample at approximately 1% containing 25 uM biotinylated rCTP, 25 uM rCTP, and 125 mM rATP, rGTP, and rUTP was used. Ampure bead cleanups were conducted at 1:1 ratios on the samples before and after the final PCR cleanup step.

## ATAC-seq experiments

Two replicates of the 4 sample/condition ATAC-seq experiments were generated on different batches of cells on different days. Cells were counted and aliquoted to 40,000 cells/tube. For second replicate, prior to the initial cell pellet collection by centrifugation, frozen drosophila S2 cells were spiked into each sample at approximately 1%. Suspension cells were then used to prepare an ATAC-seq library using the OMNI-ATAC-seq protocol [22]. We also prepared control samples for use in calling ATAC-seq peaks, genomic DNA was extracted from each of the two cell lines using the Purelink genomic mini kit (ThermoFisher, K182001), then the genomic DNA was prepared for sequencing by carrying out the OMNI-ATAC protocol on this genomic DNA suspension.

## scRNA-seq

A single replicate was prepared for single-cell RNA-seq. Samples were prepared using the 10X genomics Single Cell 3′ Reagents kit V3 (PN-1000269). Target cell number for capture was set around 3,000 cells and 40,000 reads/cell. According to the CellRanger summary output, the resulting scRNA-seq libraries contained transcripts from 2,259, 2,541, 1,653, and 2,138 cells for the four conditions (disomic control, disomic heat shock, trisomic control, trisomic heat shock respectively). The mean reads/cell for these four libraries were 49,899, 45,189, 67,070, and 56,643.

## Sequencing

PRO-seq, ATAC-seq, and scRNA-seq libraries were sequenced at the BioFrontiers Sequencing Facility (UC-Boulder). For PRO-seq experiments, samples were pooled and sequenced 1x76 single end reads, single 6 base pair index on a V2 high output 75-cycle kit for the Nextseq 500 instrument. For ATAC-seq, samples were pooled, size selected from 100–1000 base pair by blue pippen, SPRI bead cleaned up, and sequenced on a Nextseq V2 high output 75-cycle kit paired end 2x37 reads with dual 8 base pair indexes. For scRNA-seq, samples were pooled and sequenced on a Nextseq V2 High output 150 cycle kit. Samples were sequenced paired end 28 x 91, with 8 base pair index reads.

## Data analysis

**Mapping/Pipeline functions.** PRO-seq data was processed using the Nascent-Flow Next-flow pipeline version 1.3 with the following flags: –flip, –genomeid 'hg38', –singleEnd, –fstitch –dastk –tfit –saveBAM. In short, this pipeline trims reads via bbduk from bbmap version 38.05, mapped reads to GRCh38 with HISAT2 version 2.1.0 with the following flags: –very-sensitive, and –no-spliced-alignment. Sam files were converted to bam files using SAMtools version 1.8. Bedtools version v2.28.0 was used to produce bedgraph files. We checked the quality of the data via the quality control programs FastQC version v0.11.8, preseq version 2.0.3, RSeqQC version 3.0.0, and Pileup from the BBMap Suite version 38.05 were run on the data. The Nextflow pipeline Downfile pipeline version 1.0 was used to move cram files to bams, big-wigs and bedgrapahs [23]. This pipeline used samtools/1.8,bedtools/2.28.0, igvtools/2.3.75.

ATAC-seq data was processed using the Dowell lab ChIP-Flow pipeline [24] version 1.0. In short, this pipeline trimmed reads with BBDuk version 38.05 with the following settings: ktrim = r, qtrim = 10, k = 23, mink = 11, hdist = 1, maq = 10, minlen = 20. Reads were mapped to GRCh38 with HISAT2 version 2.1.0 with the following settings: –very-sensitive, –no-spliced-alignment. Sam files were converted to bam files using SAMtools version 1.8, and Picard Tools version 2.6.0 was used to identify duplicate reads and deduplicate. BEDtools version v2.28.0 was used to produce bedGraph files from bam files. IGV Tools version 2.3.75 was used to produce TDF files from bedgraphs. For quality control of libraries, the QC programs FastQC version v0.11.8, preseq version 2.0.3, RSeQC version 3.0.0, and Pileup from the BBMap Suite version 38.0 were used. The Nextflow pipeline Downfile pipeline version 1.0 was used to move cram files to bams, bigwigs and bedgrapahs [23]. This pipeline used samtools/1.8,bedtools/2.28.0, igvtools/2.3.75.

**Peak calling ATAC-seq.** HMMRATAC version 1.2.10 was used to call peaks [25]. Peaks were filtered using awk and requiring a quality score of 10 as recommended on the HMMRA-TAC github site. From the resulting gappedPeak files, only the open state regions (column 7 and 8) were used for further analysis via TFEA and via differently open peaks.

**PRO-seq identification of regions of interest.** The Nextflow pipeline Bidirectional-Flow was used to call regions of transcriptional initiation, which we call regions of interest [26]. The Bidirectional-Flow runs the Tfit algorithm on was used on each sample to annotate sites of transcriptional initiation [27]. The exact center of the Tfit regions is defined by Tfit as mu.

Mumerge [28] version 1.0 was used to combine bed files of open peak regions for all ATAC-seq data into a single bed file of active peak regions. Mumerge was used similarly to combine bed files of regions of interest for as created by Tfit for the PRO-seq data.

**DESeq2 to determine differential signal.** For the ATAC-seq, reads over the mumerged peaks were counted via featurecounts using R version 3.6.0 and the R package Rsubread 2.0.1. All genes within 25 kilobases of a differential ATAC-seq peak were used for GO analysis.

For the PRO-seq, reads over the mumerged regions of interested were counted via feature-counts using R version 3.6.0 and the R package Rsubread 2.0.1. For the PRO-seq analysis of transcription over genes, the gene locations came from the hg38 bed file. The first kb was removed from all genes so that 5′ end promoter peak in PRO-seq would not interfere with dif-ferential transcription calculation. The bed files with the 1st kb removed were saved as saf files and the saf files where used to count the number of reads over the gene bodies. Counting was done via featurecounts using R version 3.6.0 and the R package Rsubread 2.0.1 with the flag allowMultiOverlap = TRUE. For all three data sets, DESeq2 version 1.26.0 was used to deter-mine which regions/peaks/genes were differential after heat shock.

**TFEA analysis.** Transcription factor enrichment analysis (TFEA) was used to compare the relative signal in PRO-seq or ATAC-seq data nearby transcription factor motifs from one

sample to another [29]. The regions were ranked via the DESeq2 differential p-value and the direction of change (regions that were expressed higher at 42 were at the top of the file and expressed higher at 37 at the bottom of the file). We made a padded ranked file by adjusting all regions in the ranked file to be ±1500 from mu. The following TFEA dependencies were used: samtools version 1.8, bedtools 2.25.0, meme 5.0.3, R 3.6.1, python 3.6.3, matplotlib 1.5.1, scipy 0.17.1, numpy 1.14.1, htseq 0.9.1, and pybedtools 0.7.10. The basic TFEA options used were –combined file, –rank 'deseq' and the full human HOCOMOCO v10 Human meme formatted list of TFs was used for –fimo motifs. For comparisons within a cell line, the control and heat shock conditions were compared using TFEA.

For ATAC-seq, TFEA [30] was used on HMMRATAC [25] called peaks to determine which transcription factor motifs co-associate with observed changes in chromatin accessibility genome wide. TFEA calculates a Escore, or enrichment score, which measures the motif occurrence in regions of interest that have altered accessibility/transcription. For example, if heat shock causes a transcription factor to bind its motif, open chromatin, and activate transcription nearby (like HSF1), the E-score for that TF will be positive and high. On the other hand, if a transcription factor is binding to its motif, opening chromatin at 37°C and after heat shock that TF leaves DNA, then the E-score for that TF will be negative.

**TFEA plots/plot making.** For plotting the significant Transcription Factor (TF) differences between two conditions, we filtered the TFEA results by TFs that showed a corrected P value under the cutoff value, set throughout this paper to $1 \times 10^{-10}$. We then filtered out any TFs above the cutoff for significance in any of the PRO-seq TFEA comparisons between individual replicates for the same cell line/condition. We then used python version (3.6.8), jupyter version (4.4.0), and plotly version (5.8.0) to plot the scatter plot for all TFEA TF values. Plotly was also used to plot the corrected E-scores of TFs from two TFEA comparisons against each other, colored based on whether each TF had a corrected, adjusted P-value below the set cutoff threshold for neither, one, or both TFEA comparisons.

Individual gene or enhancer transcription or open chromatin were visualized using Integrative Genomics Viewer (IGV) version 2.4.10 (Figs 3B and 4A). Plotting of genome regions containing HSPB1 and SERPINH1 was done via the pygenometracks python package.

**Heatmap and metaplot production.** For making heatmaps of the data, we first worked to create bedgraphs that were normalized for depth of sequencing. For the PRO-seq data, this was accomplished via the bidirectional flow package, which calculates the millions of mapped reads via samtools [26]. For ATAC-seq, normalizing the bedgraphs to the sequencing depth was accomplished by dividing the counts in the Bedgraph of raw counts by the size factors created by DESeq2 when we determined different expressed regions of interest.

Next, HOCOMOCO was used to scan the genome for HSF1 motif locations at the p-value threshold of $1*10^{-6}$ via fimo [31]. Since most motifs are not bound, we wanted to filter the long list of all HSF1 motifs to a smaller list of regions bound by HSF1 in lymphoblastoid cells. We used the Cistrome website to find a HSF1 ChIP-seq data set from a lymphoblastoid cell line GM12878 [32] (CistromeDB record 101713), and to download the bed file of peak regions [33, 34]). We used the R packages plyranges and GenomicRanges to find all motifs within an HSF1 binding site. We also used the same packages to remove the HS1 motifs from that list that overlapped with genes when making meta-plots and heatmaps for PRO-seq. For ATAC-seq the enrichedheatmaps contained all HSF1 binding sites with HSF1 motifs regardless of if they were in a annotated gene.

We used EnrichedHeatmap to combine the normalized bedgraphs and the regions ± 500 nt from the center of the HSF1 motifs. For the heatmaps we used EnrichedHeatmap to plot regions with HSF1 motifs in an HSF1 binding site (ATAC-seq) or all HSF1 motifs in a HSF1 binding site that were not overlapping gene(PRO-seq). We used a bin size of 50 nts. The

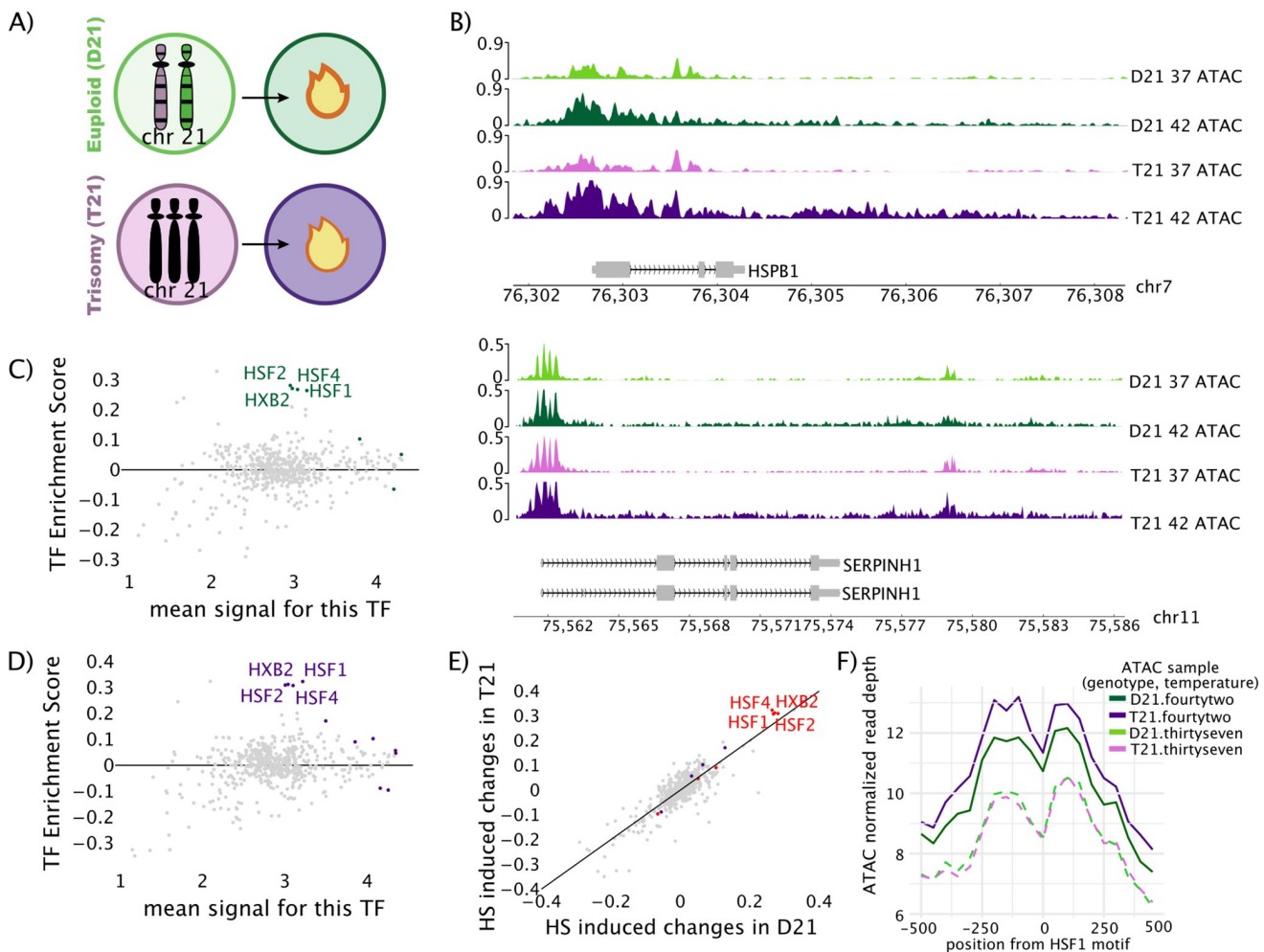

**Fig 3. Cells with trisomy 21 have increased chromatin accessibility near heat shock response elements compared to disomic controls.** A) Conceptual diagram of the conditions analyzed. Disomic cells (green) and trisomic cells (purple) at control (37˚, light color) or mild heat shock (42˚, dark color). B) ATAC-seq data traces of two heat shock regulated genes: HSPB1 (chr7:76301841–76308320) and SERPINH1 (chr11:75561294–75582876). One representative replicate is shown. Transcription factor enrichment analysis (TFEA) enrichment score (y-axis) in the C) disomic cell line and D) trisomic cell line ATAC-seq data [30]. Grey: non-significant TFs, colored: GC-corrected p-adjusted value of p<1x10⁻¹⁰. E) Scatter plot comparing heat shock-induced changes in transcription factor activity (E-values from TFEA) between disomic cells (x-axis) and trisomic cells (y-axis). Red: significant in both comparisons, Purple: significant in trisomy only, Green: significant in disomy only. All TFEA analyses (C-E) utilize both replicates. F) Averaged signal of ATAC-seq data for 1 kilobase region around active HSF1 motifs. One representative replicate is shown.

differential columns of the heatmaps were calculated by subtracting the control columns from the column with the heat shock. For the meta plots over HSF1 motifs, we calculated the colMeans over EnrichedHeatmap matrix and used ggplot2 to draw the metaplot. For the PRO-seq heatmap we used EnrichedHeatmap to plot. For the meta plot over HSF1 motifs, we calculated the colMeans over EnrichedHeatmap matrix and used ggplot2 to draw the metaplot.

**scRNA-seq mapping, normalization, and analysis.** We were given BCL files for the single-cell sequencing, which were demultiplexed and converted to fastq files using the mkfastq command in Cellranger (3.1.0). Analysis of scRNA-seq data and estimates of the cell number and average depth per cell were produced by running the Cellranger count function on the scRNA-seq data sets with the human genome (Cellranger version 3.1.0, genome data from refdata cellranger GRCh38 version 3.0.0).

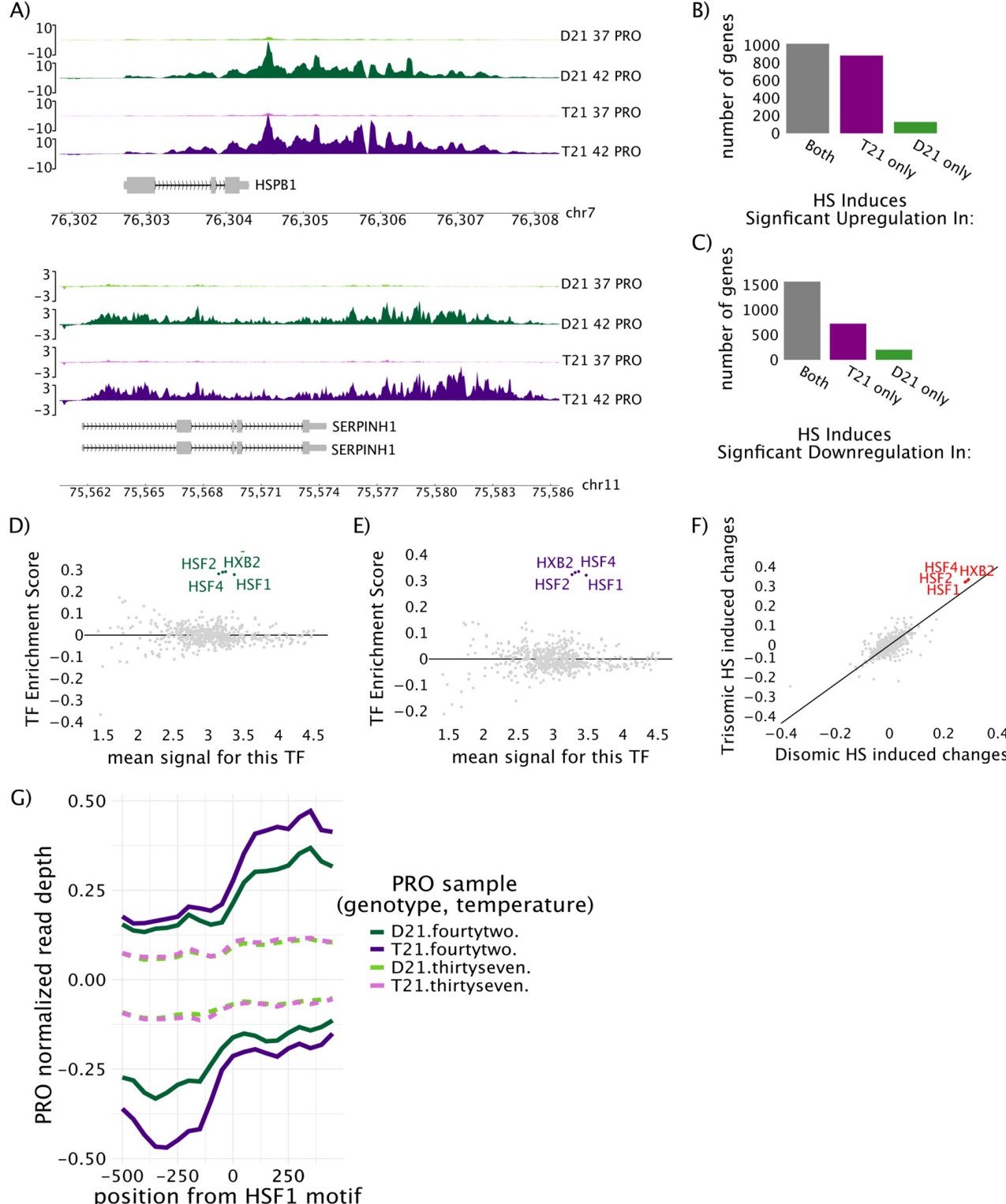

**Fig 4. A mild heat shock treatment induces more robust transcriptional changes in the trisomic cell line compared to disomic control.** A) Total normalized read count corrected PRO-seq gene traces at two heat shock regulated genes (Hspb1, Serpinh1) in the four cell types/conditions. One representative replicate is shown. Number of genes B) increasing or C) decreasing in response to heat shock (by DESeq2) in PRO-seq in one or both cell lines. TFEA enrichment score (y-axis) in the D) disomic cell line and E) trisomic cell line PRO-seq data [30]. Grey: non-significant TFs, colored: GC-corrected p-adjusted value of p<1x10$^{-10}$. F) Scatter plot comparing heat shock induced changes in transcription factor activity (E-values from

TFEA) between disomic cells (x-axis) and trisomic cells (y-axis). Red: significant in both comparisons, Purple: significant in trisomy only, Green: significant in disomy only. Panels B-F utilize both replicates. G) Average metaplot of PRO-seq data surrounding (± 500 bp) lymphoblastoid-active HSF1 motifs (at zero). One representative replicate is shown.

STARsolo version 2.7.3a was used to map the zipped fastq single cell RNA-seq data with the following options: –soloType CB_UMISimple, –soloCBWhitelist (3M-february-2018.txt.gz used from Cellranger) –soloUMIlen 12, –readFilesCommand zcat [35]. Next, the R program Matrix version 1.2–17 was used to read in the dense, filtered gene matrix output scRNA-seq matrices from Starsolo scRNA-seq into R. The scran and scater program versions 1.14.6 were used to cluster the data using the quickCluster function, calculate sum factors, and normalize the counts for each data set using the normalizeCounts function.

**scRNA-seq plot making.** The normalized counts were then used in python with plotly to make the violin plots of counts. The cell numbers were subsampled to 500 cells to make the single-cell heatmaps, and the heatmaps of the expression of each gene were plotted using plotly. A two-sample ks-test from scipy was used to determine whether each geneś counts significantly differed between samples.

**Data and code availability.** All scripts created for this publication can be found at https://github.com/maallen3/T21HSscripts/tree/master. Datasets generated during this study are deposited at the Gene Expression Omnibus (GEO): GSE173536, GSE173537.

## Results

### Individuals with trisomy 21 have elevated levels of genes related to heat shock

To determine whether heat shock response is influenced by trisomy 21, we first examined publicly available clinical data. Several projects have collected unperturbed RNA-seq data from clinical samples of multiple blood and skin cell types in individuals with and without trisomy 21, including the Human Trisome Project [36]. In the Human Trisome Project samples [1–4, 36], data accessed May 23rd, 2022), we noted that the RNA levels for several heat shock regulated genes were higher in individuals with trisomy 21, particularly in T cells and monocytes (see Fig 2A). Specifically, transcript levels for the heat shock-regulated genes HSPA8, DNAJA1, HSPH1, HSPA1A, and SERPINH1 are elevated compared to clinical controls in the trisomy 21 group. Interestingly, the transcript levels for HSF1, the master heat shock regulating transcription factor, appear to be decreased in some trisomy 21 cell types (see Fig 2B) as has been seen previously [11]. Published proteomic data confirmed that HSPA1A, HSPB1, and HSPA8 protein levels are elevated in blood samples from individuals with trisomy 21 (Fig 2C [37]). Notably, transcript and proteomic levels were not increased consistently for all HSF1-regulated genes in all blood cell types (S1 Fig). Thus, the clinical data present a perplexing picture of how trisomy 21 influences the expression of heat shock-related genes.

### Increased chromatin accessibility at HSF1 sites after heat shock in trisomic cells

To investigate the heat shock response outside of clinical complexities, we set out to characterize the acute heat shock response in paired cell lines with and without trisomy 21 (Fig 3A, Disomic cells (D21), Trisomic cells (T21)). To determine how chromatin accessibility changes as blood cells respond to heat shock, age and gender-matched lymphoblastoid cells derived from two brothers were assayed for transposase-accessible chromatin (ATAC-seq) under

control conditions 37˚C, and mild heat shock treatment. We used lymphoblastoid cells because they are a readily available blood-like cell type. A short time was chosen to focus on the primary response to heat shock, as this avoids most secondary or downstream effects arising from cellular feedback mechanisms.

We first confirmed that the heat shock response was observable under these conditions in both cell lines by manually inspecting several well-known heat shock genes, including HSPB1 and SERPINH1 (Fig 3B). These genes showed the expected response, i.e. an opening of the chromatin at the promoter region in heat shock compared to control. We called peaks in this data using HMMRATAC and determined a list of peaks that were differentially accessible after heat shock in both sets of cells. GO analysis showed that trisomy 21 and disomic samples strongly activated the heat shock pathway. Interestingly, more peaks were called differently accessible between the two conditions in the trisomic samples than in the disomic samples.

We next sought to unbiasedly infer changes in transcription factor activity (TF) in response to heat shock for both cell lines. Using transcription factor enrichment analysis (TFEA) [30] on both the disomic and trisomic cell lines, we observe that the transcription factors HSF1, HSF2, HSF4, and HXB2 were robustly induced by heat shock (Fig 3C and 3D, S3B Fig). In addition, we directly compared the Escore between the two cell lines for every TF and found that the activated TFs were consistent between the two cell lines. Notably, the activation was slightly more robust in the trisomic cells (red TFs Fig 3E).

We next sought to confirm the above result by characterizing changes in accessibility at known bound HSF1 motifs. To this end, we downloaded HSF1 ChIP-seq data from lymphoblastoid lines to identify binding sites with the HSF1 motifs [32–34] (S3A Fig). Upon heat shock, both cell lines show increases in accessibility at HSF1 sites, but the trisomic cell line has a more open ATAC-seq signal post heat shock (Fig 3F, S3A Fig). Overall, our ATAC-seq data suggests that trisomic cells display a slightly elevated chromatin accessibility at HSF1 bound sites after heat shock, compared to disomic cells. This lead us to question if the difference in chromatin was associated with concomitant changes in gene transcription in the trisomic cells upon heat shock.

## Trisomic cells have increased transcription at HSF1 motifs

To compare observed changes in chromatin accessibility to changes in transcription, we preformed precision run-on sequencing (PRO-seq) in the trisomic and disomic cells at the same time points (before and 1 hr HS) used for the ATAC-seq (Fig 3A for design). In both cell lines, heat shock genes such as HSPB1 and SERPINH1 were transcribed at higher levels after heat shock (Fig 4A, S2A Fig). We used DESeq2 to assess differential gene transcription after heat shock in both samples (Fig 4B and 4C). Many genes showed a reduction in transcription in response to heat shock in both cell lines (Fig 4C, S2B Fig). The trisomic sample revealed more genes with significant changes in transcription than the disomic sample (Fig 4B and 4C). Moreover, genes that were differentially transcribed in both samples showed a general trend of being induced to a greater extent in the trisomic cell line (S2B and S2C Fig).

To determine if HSF1 was the only TF with increased transcription associated with its motifs, we used TFEA to infer transcription factor activity changes based on the PRO-seq data (independent of the changes inferred from ATAC-seq). To this end, we used Tfit (Transcription fit) to identify all sites of bidirectional transcription within each PRO-seq data set [38]. Regions of transcription were combined across conditions and replicates using muMerge [30]. Consistent with the ATAC results, TFEA shows a robust activation of HSF1, 2, and 4 in PRO-seq in both the disomic and trisomic cell line (Fig 4D and 4E). A direct comparison of the heat

shock induced changes in TF activity revealed a higher relative activation of HSF TFs in the trisomic cell line compared to the disomic cell line (Fig 4F).

Since transcription factor binding sites co-occur with enhancer RNAs which are readily detected by the PRO-seq assay, we next examined nascent transcription at HSF1 binding sites in response to heat shock. We hypothesized that a more sensitive or robust HSF1 activation in the trisomic cells might explain the increased genome wide changes in chromatin accessibility and transcription in the trisomic cell line compared to the disomic line. Genome wide, heat shock led to an increase in PRO-seq signal at bound HSF1 motifs in both trisomic and disomic cell lines, confirming the activation of HSF1 motif adjacent eRNAs in both cell lines (S3C Fig, Fig 4G). Though PRO-seq levels began at similar levels in the two cell lines under control conditions, after one hour of mild heat shock treatment we noted a more robust transcriptional response in the trisomic cell line compared to the disomic cell line (Fig 4G). Collectively, both the PRO-seq and the ATAC-seq suggest that though the transcriptional response to heat shock is similar between the two cell lines, but more robust in the trisomic cells.

## Single cell RNA sequencing confirms effect is population wide

In yeast, aneuploidy increases cell to cell variation in response to heat shock [20]. Therefore, we next asked whether the increase in heat shock response observed in trisomic cells arises from a small number of hyper-responsive cells or a more consistent population wide effect. To address this question, we applied single-cell RNA sequencing (scRNA-seq) to the same cell lines, both before and after the 1 hr mild heat shock. We reasoned that if a small population of cells had an unusually strong heat shock response, those cells should be expected to have signals exceedingly higher than the other cells. Furthermore, the non-outlier cells would be expected to have expression levels roughly equal to that of the disomic cells. On the other hand, if the whole population of trisomic cells showed a more robust heat shock than the entire population of disomic cells, the distribution of cell sums would merely be shifted.

Transcription levels are expected to follow DNA dosage [39–41]. Therefore, we first confirmed that the scRNA-seq protocol detected the expected higher quantity of chromosome 21 transcripts in the trisomy 21 sample, as a control. As expected, when we plotted the depth normalized counts per cell for chromosome 21 encoded genes we found that transcripts on chromosome 21 are present in higher quantities in the trisomic cell line than the disomic cell line (S4A and S4B Fig). Additionally, we examined the transcript levels for known heat shock-responsive genes and confirmed heat shock-induced increases in the transcript level for these genes in both cell lines (Fig 5C and 5D).

To address whether a small number of outlier cells drove the observed increase in trisomy heat shock response, we summed all Z-scores for all heat shock genes across individual cells. The results show a shift in the distribution of cell sums, indicating the increase in expression is population wide (Fig 5E). Essentially all trisomic cells appear to have an increase in heat shock transcripts relative to the disomic cells. The same result is obtained when using the median of Z-scores rather than the mean (Fig 5F). Importantly, while some cells respond more strongly than others to heat shock, the T21 cells do not show an overall pattern different from the D21 cells, consistent with a population-wide increase not arising from a small number of hyper-responding cells.

## Discussion

In a pair of EBV immortalized lymphoblastoid cell lines derived from brothers, with and without Down syndrome, we observed that the T21 cells respond more robustly to mild heat shock (Fig 1). This increase was observed in immediate (1 hr) transcription response, accessibility

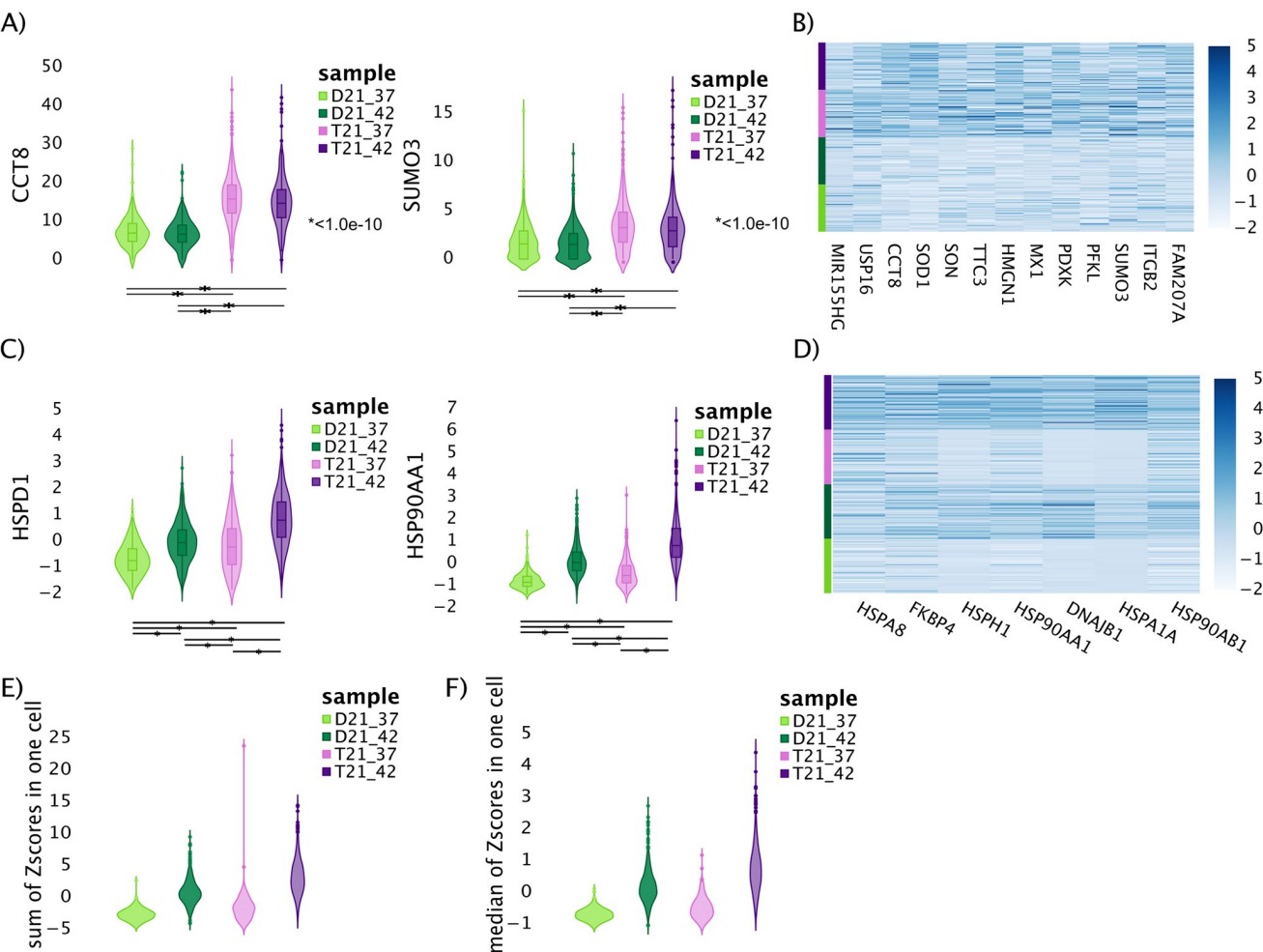

**Fig 5. Single Cell RNA-seq indicates that changes in heat shock-induced gene expression in trisomy 21 cells are population-wide.** A) Violin plots of percent of total normalized scRNA-seq gene counts per cell for two genes (CCT8, SUMO3) present on chromosome 21 and non-responsive to heat shock. B) Heatmap of the Z-scores of chromosome 21 genes showing increased reads in trisomy 21 cells. C) Violin plots of two heat shock responsive genes (HSPD1, HSP90AA1) not encoded on chromosome 21; the y-axis is the levels across more than 500 cells in each sample via scRNA-seq. D) Heatmap of the Z-scores of heat shock genes shows a general up-regulation in the expression of heat shock genes, rather than a few cells with extreme heat shock phenotypes. Violin plots showing the E) sum and F) median of Z-scores of all heat shock genes for more than 500 cells in each sample. The genes showed must be present in at least 75% of cells. All panels: control (light color) and heat shock (dark color) of disomic (green) and trisomic (purple) lymphoblastoid cells.

changes, and in HSF regulated steady-state transcript levels. However, given that we only examined a single pair of cell lines, it remains unclear whether the effect was truly induced by trisomy 21 or is some unique property of these individuals or these cell lines. More individuals are needed to resolve this ambiguity. If the extension to additional individuals were to include alternative immortalization and/or primary cells, we would also be able to confirm that this result is not influenced by the EBV immortalization strategy. Moreover, extensive clinical analysis could determine whether the misregulation of HSF1 is a hallmark of Down syndrome.

As to the origins of the dysregulation, three genes with links to the heat shock response are present on chromosome 21: HSPA13, DNAJC28 and HSF2BP. It is possible that one of these genes may be impacting the heat shock response directly (Fig 1B). However, there are no known mechanisms linking these genes to the regulation of the heat shock response.

Furthermore, the low-level expression of these genes in lymphoblastoids would make a causal role in increased heat response a surprising mechanism.

An alternative explanation for the elevated heat shock response is stress response pathway cross-talk. While HSF1 activation is generally via heat shock, it can also occur as a result of heat shock independent stresses such as proteotoxic stress from ribosomal gene imbalances [42]. Chromosome 21 encoded genes such as SOD1, DYRK1A, and the four interferon response receptors are directly involved in cellular stress response pathways. Furthermore, previous studies have suggested that these genes may lead to the over-activation of stress response pathways in individuals with trisomy 21 [1–7]. Consistent with this idea, we do see that the interferon receptor genes are overactivated in the trisomic lymphoblastoid cell line in our data (S3C Fig). Cross-talk between these stress responses may cause the overactivated heat shock response observed in this study and might impact other cellular responses to perturbations. Alternatively, the presence of an extra-chromosome, regardless of the identity of the chromosome has also been shown to induce stress phenotypes [8–12, 39].

Despite observing increased heat shock induced HSF1 activity in the human trisome project, we did not detect a difference between the two cell lines in the transcription levels of the HSF1 transcript itself by PRO-seq (adjusted p-value 0.9999577676184). However, HSF1 is a highly regulated TF [13–16] and some mechanisms of elevated HSF1 activation are downstream of transcript levels. For example, DAXX, TPR, Mediator, and ribosomal components are all genes known to regulate HSF1 activity [14, 15, 42]. Indeed, transcript levels for many of these factors are elevated in our scRNA-seq data (S4 Fig) and in some cell types of individuals within the human trisomy project data (Fig 2).

Finally, it is worth noting that a misregulation of heat shock has potential consequences. An irregular response to heat shock could hamper the ability of the immune system to respond to infection in the presence of a fever. Further, if blood cells with Trisomy 21 over-respond to many common cellular stresses, then individuals with Down syndrome may be in a constant state of stress. Admittedly, measuring the levels of stress response relevant genes in clinical samples could be complicated by the presence of a host of co-morbidity conditions, but these stresses could also be contributing to the co-morbid conditions. The observation here that a lymphoblastoid cell line with trisomy 21 over-responded to a stress not known to be regulated by a chromosome 21 gene further emphasizes the important role of regulatory networks in Down syndrome.

## Conclusion

We found that the presence of a third copy of chromosome 21 did not disrupt the cellular ability to mount a heat shock response. Rather, we observed that the trisomic cells were surprisingly agile at changing gene expression in response to this perturbation and appeared to increase both chromatin accessibility and transcription at HSF1 motifs more readily than the disomic control. Our global analysis of changes in chromatin accessibility and nascent transcription, found that the trisomic cells responded more aggressively to the mild heat shock treatment. After one hour at 42°C, we found a more robust activation of HSF1 activity in trisomic cells as inferred from the magnitude of changes in ATAC-seq data, PRO-seq data, and HSF1 regulated steady-state transcript levels in scRNA-seq data.

## Supporting information

**S1 Fig. Heat shock genes altered in transcript or protein levels.** All data from the Human Trisome Project. A) Heat shock relevant genes (DNAJC28, DNAJB1, HSPA13, SOD1, DAXX, HSF4), are elevated (RNA-seq) in individuals with trisomy 21 (purple) relative to disomic

controls (green). B) Proteomic data for HSP90B1 which shows increased protein levels in individuals with trisomy 21.
(TIF)

**S2 Fig. Extended heatmaps of ATAC-seq and PRO-seq signal over HSF1 sites.** A) Heatmap of ATAC-seq data surrounding regions identified from HSF1 ChIPseq peaks with HSF1 motif instances. First four columns (in red) show DESeq2 size factor normalized ATAC-seq data at (± 500 nts) for 589 regions. The fifth and sixth columns (in black and orange) show the differential signal (heat shock—control). Top of the heat map is meta plot of the mean signal per position. One representative replicate is shown. B) A plot showing TFs reported as significantly changed by TFEA in at least one comparison. Diamonds: statically significant (p-value $<1\text{x}10^{-10}$), Circles: not significant. Left two columns show the TFEA E-score for ATAC-seq after heat shock, whereas the right two columns show TFEA E-score for PRO-seq. E-score, or enrichment score, measures the motif co-occurrence with open chromatin (ATAC) or transcription (PRO). All TFEA analyses utilize two replicates. C) Heat maps of intergenic HSF1 bound regions (identified in ChIP-seq, requires motif) for ATAC-seq and PRO-seq data. Columns correspond to: differential ATAC-seq (black/orange, columns 1–2), ATAC-seq signal (red, columns 3–6), differential PRO-seq signal (black/orange, columns 7–10), reverse strand PRO-seq (red, columns 11–14), forward strand PRO-seq (blue, columns 16–18). Top: line graph of mean depth per position. All columns are centered at the HSF1 motif, dashed line.
(TIF)

**S3 Fig. Extended plots of PRO-seq gene transcription.** A) Volcano plots of differentially transcribed genes after heat shock. Left: Disomy, Right: Trisomy. Green: up regulated after heat shock, Red: down regulated. B) Scatter plot of log fold change observed for statistically significant differences with heat shock showing T21 (x-axis) and D21 (y-axis) samples. Colored by condition in which the significance call was made: T21 (purple), D21 (green) or both (grey). Best fit line is relative to set of grey genes. C) A bar graph of the log fold change of Heat shock genes (as defined by the GO term HEAT_SHOCK_PROTEIN_BINDING. The top plot contains the heat shock genes changed only in the trisomic sample. The bottom plot contains heat shock genes differently expressed in both cell types.
(TIF)

**S4 Fig. Extended plots of scRNAseq.** A) The same genes as in Fig 4A but with Z scores instead of raw counts. B) The same genes as in Fig 4C but with raw counts instead of Z scores.
(TIF)

## Acknowledgments

We would like to thank the employees of the Translational Nexus Biobank (COMIRB 08-1276), the University of Colorado School of Medicine, JFK Partners (Angela Rachubinski, Gessi Pino, Karl Pfenninger, Holly Sullivan, Cordelia Robinson Rosenberg). We acknowledge the BioFrontiers Computing Center at the University of Colorado Boulder for providing High Performance Computing resources supported by BioFrontiers Information Technology. We would also like to thank the BioFrontiers Information Technology for there generous support. We would like to thank the lab of Rui Yi lab, and specifically Dr. Dongmei Wang, for the use of their 10X instrument and their support in producing the single-cell libraries. We would also like to thank the Human Trisome Project for making their data easy to use and open source.

## Author Contributions

**Conceptualization:** Joseph F. Cardiello, Robin Dowell, Mary Ann Allen.

**Data curation:** Joseph F. Cardiello, Jessica Westfall, Robin Dowell, Mary Ann Allen.

**Formal analysis:** Joseph F. Cardiello, Mary Ann Allen.

**Funding acquisition:** Robin Dowell, Mary Ann Allen.

**Investigation:** Joseph F. Cardiello, Mary Ann Allen.

**Methodology:** Joseph F. Cardiello, Robin Dowell, Mary Ann Allen.

**Project administration:** Robin Dowell, Mary Ann Allen.

**Resources:** Mary Ann Allen.

**Software:** Joseph F. Cardiello.

**Supervision:** Robin Dowell, Mary Ann Allen.

**Validation:** Joseph F. Cardiello, Mary Ann Allen.

**Visualization:** Joseph F. Cardiello, Jessica Westfall, Mary Ann Allen.

**Writing – original draft:** Joseph F. Cardiello, Robin Dowell, Mary Ann Allen.

**Writing – review & editing:** Joseph F. Cardiello, Jessica Westfall, Robin Dowell, Mary Ann Allen.

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
