## [Decision Letter · Decision Letter 0]

3 Nov 2023

PONE-D-23-19543Characterizing Primary transcriptional responses to short term heat shock in paired fraternal lymphoblastoid lines with and without Down syndrome.PLOS ONE

Dear Dr. Allen,

Thank you for submitting your manuscript to PLOS ONE. After careful consideration, we feel that it has merit but does not fully meet PLOS ONE’s publication criteria as it currently stands. Therefore, we invite you to submit a revised version of the manuscript that addresses the points raised during the review process. Two reviewers have commented on your manuscript. The feedback they have provided differs, but both provide insights on how to make the manuscript better. I agree that the manuscript needs improvement. I acknowledge that you may disagree with some of the comments (for example, about changing the title). Thus, I can give you some discretion on whether you make some of the changes. However, be sure to provide strong justification for any change that you choose not to make. Thank you for sharing the code on GitHub. Please provide a clear README file that explains to others how to execute the analysis after they have downloaded your repository.

We look forward to receiving your revised manuscript.

Kind regards,

Stephen R. Piccolo

Academic Editor

PLOS ONE

Journal Requirements:

2. You indicated that ethical approval was not necessary for your study. We understand that the framework for ethical oversight requirements for studies of this type may differ depending on the setting and we would appreciate some further clarification regarding your research. Could you please provide further details on why your study is exempt from the need for approval and confirmation from your institutional review board or research ethics committee (e.g., in the form of a letter or email correspondence) that ethics review was not necessary for this study? Please include a copy of the correspondence as an ""Other"" file.

"We would like to acknowledge funding from the Sie Foundation for funding JFC and MAA, and for funding from the R01HL156475 MAA and JW, and R01GM125871 for funding MAA and RDD."

6. Thank you for stating the following in the Competing Interests section: 

"RDD and MAA have a patent for “Methods for predicting transcription factor activity” that is not directly related to the work contained in this data note."

Reviewers' comments:

Reviewer's Responses to Questions

**Comments to the Author**

1. Is the manuscript technically sound, and do the data support the conclusions?

Reviewer #1: Yes

Reviewer #2: Yes

2. Has the statistical analysis been performed appropriately and rigorously? 

Reviewer #1: No

Reviewer #2: Yes

3. Have the authors made all data underlying the findings in their manuscript fully available?

Reviewer #1: Yes

Reviewer #2: No

4. Is the manuscript presented in an intelligible fashion and written in standard English?

Reviewer #1: Yes

Reviewer #2: Yes

5. Review Comments to the Author

Reviewer #1: In this work authors present evidences of specific heat shock signaling activation upon presence of an extra chr. 21, which is not restricted to the chr. 21 gene expression responses and rather illustrates the existence of the chronic stress phenotype. This paper adds an important peace to the puzzle, however there are suggestion below to improve the manuscript.

Mainly, the writing of the paper is not very good. It is too wordy and difficult to read. Many things are repeated again and again.

Abstract should be rewritten. It is too wordy, and phrasing is not the best. Scientific English is concise and “to the point”.

Title of the article is too long and contains unnecessary information for the main message. I suggest to change the title.

Introduction

Lines 9-13. Very long and badly phrased sentence. It is already known (and not “unclear” as state the authors) that the responses from the presence of extra chr. 21 are not only restricted to the increased expression of genes encoded on chr. 21. The following references should be introduced PMID: 34215848, PMID: 36459979, PMID: 36496311, PMID: 27308438, PMID: 25205676

Materials and Methods

Lines 43-55 should be moved to the results. Also contains repetitive info e.g. about Nexus biobank (lines 43 and 57).

Line 64. The cells were kept in waterbath at 42 degrees for 1h. So they were kept out of incubator and therefore, lacked CO2 for 1h? Can authors exclude the damaging effect from lack of CO2? If there’s any? And separate it from the heat shock?

Results

The figures labeling does not correspond reality. Please revise.

Lines 256-258, line 261 HSF1 levels was not “changes” and was not “not increased” but was “decreased” in certain conditions. This was also shown by Donnelly et al., 2014.

Lines 272-274 Repeated sentence.

Line 271, 274. You don’t have to repeat all the time “1 hour”.

Lines 287-293 Details more suitable for Materials and Methods section.

Line 317. Words are missing in the sentence.

Figures 3-5. The two shades of green are very similar and difficult to distinguish on the graphs.

Lines 358-360 The title is too long.

P values should be indicated on the graphs e.g. of Figure 5

Reviewer #2: In this manuscript, Cardiello et al explore heat shock related genes from public datasets and EBV transformed lymphoblastoid cell lines they derive from siblings with and without trisomy 21 (T21). A careful and controlled preliminary analysis is undertaken using clear bioinformatic workflows for molecular assays to identify transcriptional differences between the cell lines where T21 gene dose is a covariate that may be influencing gene expression in the context of heat-shock stress.

Assessing public available RNAseq and proteomic data on a number of tissues from Down syndrome (Human Trisome Project), early gene expression with PROseq and ATACseq for chromatin accessibility, and scRNAseq approaches comparing the lymphoblastoid cell lines, the authors conclude:

1. a number of Heat shock genes are inconsistently upregulated in T21 cohorts compared to disomic controls, including HSF1, a heat shock regulating transcription factor, from publicly available datasets

2. comparison between the lymphblastoid cell lines (T21 and disomic control) show

- increased chromatin accessibilty at specific heat shock response elements (HSF1, SERPINH1) with heat shock stress with an inferred increased in TF activity using Transcription Factor Enrichment Analysis of ATACseq data, and further (and perhaps difficult to interpret) analysis undertaken for sites enriched for one HSP TF, HSF1 binding by ChIPseq (Fig S3)

3. Heat shock response was associated with greater differentially transcribed genes (up and down) in T21 cell line with enrichment in heat shock associated TFs (HSF1, HSF2, HSF4, HXB2), with PRO-seq suggesting "more robust" nascent RNA transcription in T21 cell line with heat shock.

4. Clonal heterogeneity in the cell lines with heat shock using scRNAseq was examined, demonstrating that at least for some certain heat shock responsive genes, changes in transcription appear at some degree to be at a "population" level, rather than dramaticly responding outlier clones.

The authors appropriately state that further studies are required to understand the implications of these preliminary, albeit convincing, observations in blood and particularly, immune cells.

Major comments

1. Much of the detailed molecular characterisation of heat shock response is conducted on disomic and T21 lymphoblastoid cell lines from (I assume) genetically related brothers that had been immortalised by identical batch EBV transduction.

Could the authors comment about the generalisability of their observations for these "single" replicate analyses. Are there other available sib pairs (disomic and trisomic) for instance in the Nexus biobank to be able to undertake replicate analysis of at least some of the assays? If not, is there benefit/possibility of deriving and underaking a subset of analyses (eg. RNAseq) from transduction of the blood cells with another EBV transduction (as pseudo-biological replicates)? If not, I think these observations need to be qualified in this context by the authors.

2. While perhaps inconsistently described in the methods, replicates undertaken for each assay, particularly using the lymphoblastoid cell lines, should be described in the legend.

3. The specific public dataset data used for analysis from the Human Trisome Project, in this manuscript, should be provided (eg. which coded sample identifiers for each sample) eg as a supplementary table for reproducibility.

4. Fig 3F is missing a legend

5. Supp Fig 3C: Differential ATACseq and PROSeq data requires some clarity with regard to the visual plots including the directionality of the data (the conditions appear to be missing for each column (cf Fig S3A) and what each column represents (?technical replicates I am assuming - see comment 2)

6. missing "to" in "next sought (to) confirm" line 306 page 11

6. PLOS authors have the option to publish the peer review history of their article (what does this mean?). If published, this will include your full peer review and any attached files.

Reviewer #1: No

Reviewer #2: No

---

## [Author Response · Author response to Decision Letter 0]

12 Mar 2024

The response to reviewers is in the attached PDF.

---

## [Decision Letter · Decision Letter 1]

11 Apr 2024

PONE-D-23-19543R1Characterizing primary transcriptional responses to short term heat shock in Down syndrome.PLOS ONE

Dear Dr. Allen,

Thank you for submitting your manuscript to PLOS ONE. After careful consideration, we feel that it has merit but does not fully meet PLOS ONE’s publication criteria as it currently stands. Therefore, we invite you to submit a revised version of the manuscript that addresses the points raised during the review process.

We look forward to receiving your revised manuscript.

Kind regards,

Stephen R. Piccolo

Academic Editor

PLOS ONE

Journal Requirements:

Additional Editor Comments:

Both reviewers have examined the revised manuscript and provided brief comments. Please address the suggestion given by the first reviewer regarding statistical significance. Then resubmit.

Reviewers' comments:

Reviewer's Responses to Questions

**Comments to the Author**

1. If the authors have adequately addressed your comments raised in a previous round of review and you feel that this manuscript is now acceptable for publication, you may indicate that here to bypass the “Comments to the Author” section, enter your conflict of interest statement in the “Confidential to Editor” section, and submit your "Accept" recommendation.

Reviewer #1: All comments have been addressed

Reviewer #2: All comments have been addressed

2. Is the manuscript technically sound, and do the data support the conclusions?

Reviewer #1: Yes

Reviewer #2: Yes

3. Has the statistical analysis been performed appropriately and rigorously? 

Reviewer #1: Yes

Reviewer #2: Yes

4. Have the authors made all data underlying the findings in their manuscript fully available?

Reviewer #1: Yes

Reviewer #2: Yes

5. Is the manuscript presented in an intelligible fashion and written in standard English?

Reviewer #1: Yes

Reviewer #2: Yes

6. Review Comments to the Author

Reviewer #1: There are extra brackets for the references in the introduction part that should be removed. Also if changes in the graphs are not statistically significant you should put "NS" on the graph. Please look through your conclusions again because if there's NS change you can not say that there is an increase or decrease as you state e.g. in Fig. 5.

For the rest paper is adequately respond to the requirement and can be therefore accepted for publication.

Reviewer #2: I have reviewed the resubmission. I thank the authors for addressing my comments I had made on their original submission.

7. PLOS authors have the option to publish the peer review history of their article (what does this mean?). If published, this will include your full peer review and any attached files.

Reviewer #1: No

Reviewer #2: **Yes: **Ashley P Ng

---

## [Author Response · Author response to Decision Letter 1]

2 Jul 2024

All reviewer responses are in the attached file. P-values have been added to the figures as requested.

---

## [Editor Report · Decision Letter 2]

4 Jul 2024

Characterizing primary transcriptional responses to short term heat shock in Down syndrome.

PONE-D-23-19543R2

Dear Dr. Allen,

We’re pleased to inform you that your manuscript has been judged scientifically suitable for publication and will be formally accepted for publication once it meets all outstanding technical requirements.

Kind regards,

Stephen R. Piccolo

Academic Editor

PLOS ONE

Additional Editor Comments (optional):

Thank you for addressing the reviewers' comments and thereby improving the manuscript.
---

## [Editor Report · Acceptance letter]

29 Jul 2024

PONE-D-23-19543R2 

PLOS ONE

Dear Dr. Allen, 

I'm pleased to inform you that your manuscript has been deemed suitable for publication in PLOS ONE. Congratulations! Your manuscript is now being handed over to our production team.

Kind regards, 

on behalf of

Dr. Stephen R. Piccolo 

Academic Editor

PLOS ONE